



# Planetary Radar Science Case for EISCAT 3D

Torbjørn Tveito[1], Juha Vierinen[1], Björn Gustavsson[1], and Viswanathan Lakshmi Narayanan[1]

[1]University of Tromsø, The Arctic University of Norway, Postboks 6050 Langnes, 9037 Tromsø, Norway

**Correspondence:** Torbjørn Tveito (torbjorn.tveito@uit.no)

**Abstract.** Ground-based inverse synthetic aperture radar is a tool that can provide insights into the early history and formative processes of planetary bodies in the inner Solar System. This information is gathered by measuring the scattering matrix of the target body, providing composite information about the physical structure and chemical makeup of its surface and subsurface down to the penetration depth of the radio wave. This work describes the technical capabilities of the upcoming
233 MHz EISCAT 3D radar facility for measuring planetary surfaces. Estimates of the achievable signal-to-noise ratios for terrestrial target bodies are provided. While Venus and Mars can possibly be detected, only the Moon is found to have sufficient signal-to-noise ratio to allow high resolution mapping to be performed. The performance of the EISCAT 3D antenna layout is evaluated for interferometric range-Doppler disambiguation and it is found to be well suited for this task, providing up to 20 dB of separation between Doppler north and south hemispheres in our case study. The low frequency used by EISCAT 3D is
more affected by the ionosphere than higher frequency radars. The magnitude of the Doppler broadening due to ionospheric propagation effects associated with traveling ionospheric disturbances has been estimated. The effect is found to be significant, but not severe enough to prevent high resolution imaging. A survey of Lunar observing opportunities between 2022 and 2040 are evaluated by investigating the path of the sub-radar point when the Moon is above the local radar horizon. During this time, a good variety of look directions and Doppler equator directions are found, with observations opportunities available for
approximately ten days every lunar month. EISCAT 3D will be able to provide new high quality polarimetric scattering map of the near-side of the Moon with a previously unused wavelength of 1.3 m, which provides a good compromise between radio wave penetration depth and Doppler resolution.

## 1 Introduction

Ground based radio remote sensing of planetary surfaces is a remote sensing technique wherein a planetary target is illuminated
by a radar transmitter on the surface of Earth. This is a tool that can provide insights into the formative processes of the surfaces of the inner planets, as well as the conditions in the early history of the Solar System (Ostro, 1993; Campbell, 2016). Radar mapping of planetary surfaces provides information about the properties of the upper layers of the planet's surface, and can also provide accurate information about an object's rotation (Campbell et al., 2019). Actively transmitting the radio wave allows researchers to control the frequency, amplitude, and polarization of the illuminating beam. This means that we can extract
information about the surface from the ways it affects these parameters. Remote sensing techniques are often an inexpensive, less resource intensive alternative to in-situ measurements to obtain information about large swathes of planetary surfaces.



It is impossible to summarize here all of the scientific highlights of previous planetary radar research. However, we have tried to list topics that may be of interest to study using EISCAT 3D. Previous lunar radio remote sensing studies have been used to determine the average roughness of the lunar surface on several length scales, as well as investigate the geology of

the upper crust of the lunar nearside (Evans and Hagfors, 1968). Radar studies of planetary targets in the Solar System have been used to look for water ice in permanently shadowed craters (Stacy et al., 1997). These craters tend to be near the poles, where the crater rims shields the interior from sunlight. Spudis et al. (2013), using the Mini-RF instrument onboard the Lunar Reconnaissance Orbiter in conjunction with the Arecibo radar, found evidence of such water deposits near the lunar poles. Radar-bright features consistent with water ice have been found on Mercury's permanently shadowed polar craters (Harmon

et al., 2011) using only the Arecibo radar. Radar studies have also been used to map the surface of Venus through its visually-opaque clouds, as well as determine its retrograde rotation (Goldstein and Carpenter, 1963). These maps revealed a surface dominated by volcanism, with a large amount of shield volcanoes and relatively few craters. This suggested that the planet was geologically active relatively recently, and may still be. A continuing radar campaign aimed at looking for changes in the Venusian landscape could give information regarding the temporal and spatial scale of geologic activity (Shalygin et al., 2012).

The use of many different wavelengths to investigate planetary surfaces is important in order to build a more complete understanding of the properties of the target body (Thompson, 1979). This is because the radio wave is modified by spatial variations in the dielectric constant of the medium it interacts with. These variations can be caused by both structural and chemical variations in the medium. Specular scattering is dominated by smooth surfaces oriented normally to the radar line of sight. Structural variations on the order of $10\%$ to $100\%$ of the wavelength contribute the majority of the depolarized

component of the reflected signal. Simultaneously, the wave will attenuate when it penetrates the target body. The penetration depth is the point where the power of the wave is reduced to $e^{-1}$, and is roughly proportional to the wavelength used to investigate (Campbell, 2002). These two effects, in tandem, mean that different wavelengths provide different, complimentary slices of information. Another parameter that is affected by the frequency of the illuminating wave is the Doppler resolution achievable. Longer wavelengths are not able to obtain as fine a resolution as shorter wavelengths if observation times are the

same.

There are several different types of ground-based radar remote sensing experiments of planetary targets. Range-Doppler mapping of radio albedo can provide information about the structural and dielectric properties of the surface of the body being observed. If polarimetric information is available, the near subsurface can be partially separated from the surface echo, providing more information about the subsurface structure and chemical composition. If several receivers are available, topographical

mapping is possible by using the phase difference between the received signals to calculate differences in optical path-length (Margot et al., 2000). This type of interferometric calculation can also be used to resolve ambiguities in range-Doppler mapping caused by target geometries (Rogers and Ingalls, 1969). Radar measurements can also be used to ascertain the distance to an object, and its relative velocity. Repeated measurements can then help constrain the orbital characteristics of Near Earth Objects (NEOs), identifying potentially hazardous space objects (Giorgini et al., 2008). Radar observations of meteoroids can

also aid in determining its spin state through investigating Doppler spectra or radar speckle patterns (Busch et al., 2010). A





detailed analysis of the capabilities of EISCAT 3D in relation to observation of NEOs has been done in a companion paper by Kastinen et al. (2020).

Previous long-wavelength ground based inverse synthetic aperture radar maps of the lunar surface include a map of backscatter at 40 MHz (7.5 m) by Thompson (1978). This observation provided the first high-resolution map of the lunar nearside for

long-wavelength radar, as well as a measurement of the average scattering properties of the lunar regolith at long wavelengths. A recent study by Vierinen et al. (2017b) provided a view of the lunar nearside at 6 meters, where both the specular and orthogonal-to-specular polarizations were recorded. This gives a view of the properties of both the surface and subsurface. The long wavelength studies have identified two regions, 1) the Schiller-Zucchius basin and 2) the highlands around Montes Jura, which have anomalously low depolarized radar returns (Thompson et al., 2006; Vierinen et al., 2017b). The use of 230 MHz

radar maps would be of interest in order to constrain the physical mechanism that causes this reduced return.

In this paper, we discuss the upcoming EISCAT 3D radar facility and its capabilities in the context of planetary radar studies. In section 2, we describe the performance parameters of the radar. Section 3 investigates the detectability of Mars, Venus and Mercury, as well as the Moon. In section 4, we discuss an interferometric technique for disambiguating Doppler North and South from one another, and estimate the achievable contrast obtainable using the EISCAT 3D interferometer. In Section 5, we

investigate how severely ionospheric radio propagation will degrade the Doppler resolution radar maps. This is done using a a first order model for traveling ionospheric disturbances. Section 6 outlines Lunar observing opportunities between 2022 and 2040 using EISCAT 3D.

## 2   EISCAT 3D

EISCAT 3D is a new multi-static high power large aperture phased array radar, scheduled for first-light experiments in 2022.

The facility is currently being built in northern Fennoscandia, with the transmitter located in Skibotn, Norway, as can be seen in figure 1. The primary science case for this new facility is ionospheric and upper atmospheric physics (McCrea et al., 2015). However, this radar will also potentially be very useful for planetary radar studies.

The radar facility will be able to transmit and receive signals at elevations down to $30°$ above the horizon. This will allow both a view of the ecliptic plane and lunar observations of several hours. The facility will operate with a 233 MHz center

frequency, a 5 MHz transmit bandwidth, a 30 MHz receive bandwidth (Vierinen et al., 2017a), a system noise temperature of 150 K, a peak transmit power of 5 MW, and a maximum duty-cycle of 25% (Kero et al., 2019). The main antenna arrays have a maximal diameter of 75 m, and a maximal gain, $G_0$, of 43 dB towards zenith. The decline in gain as a function of angle of incidence can be approximated as the reduction in projected area, $G = G_0 \cos\theta$. The radar allows independent transmit and receive on two orthogonal linear polarizations. As a consequence of this, any polarization state vector can be synthesized. The

wavelength has not, to our knowledge, been used previously for lunar studies. This means that the radar will be able to sample a new scale-size in lunar surface roughness and subsurface structure.

Another benefit of the EISCAT 3D facility is the availability of numerous interferometric baselines, with distances over 100 km possible. See figure 1 for the planned configuration of antenna placement. The Skibotn antenna location will have





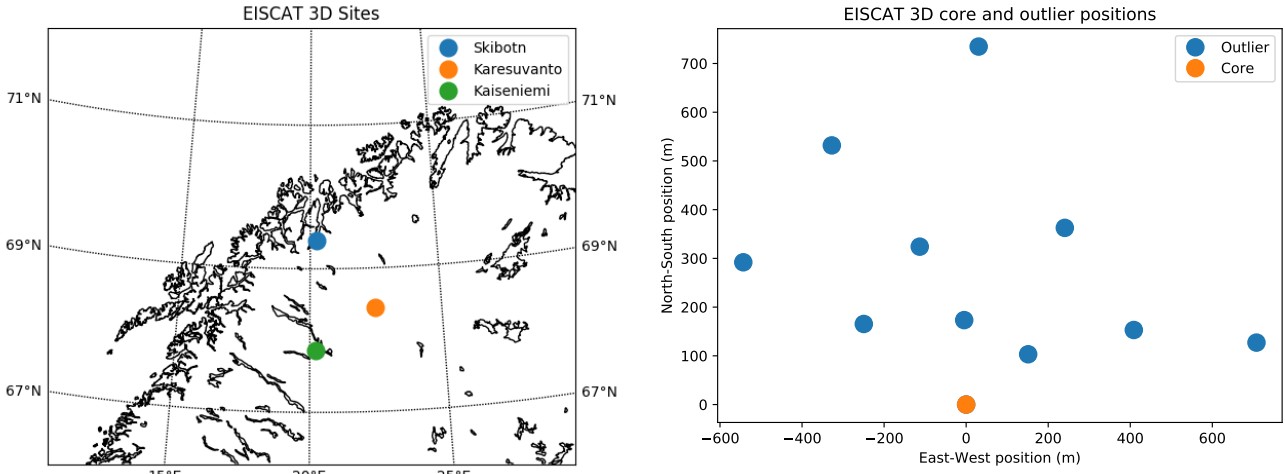

**Figure 1.** Left: Locations of the three receiver sites of EISCAT 3D. The Skibotn site has both transmitting and receiving capabilities, while the Kaiseniemi and Karesuvanto are only able to receive. Right: planned distribution of antenna modules at the Skibotn site. Note that only the Skibotn site is planned to have outlier antenna modules intended for interferometric purposes.

ten outlier antenna modules, each with a maximal diameter of approximately 7.9 m consisting of 91 dipole antenna elements.
While these additional antenna modules don't provide as high a signal-to-noise ratio, they can be useful in providing a large number of interferometric baselines. In total, when using all three receiving sites and the Skibotn outlier antennae, one can create 78 unique antenna pairs.

## 3   Detectability of planetary bodies

In this section, we evaluate the signal-to-noise ratios (SNRs) of planetary bodies when observed by EISCAT 3D. The expression
for the signal-to-noise ratio for a planetary target is given by Ostro (1993). These expressions are also derived and discussed in a companion paper by Kastinen et al. (2020), in the context of detectability of Near-Earth Objects using the EISCAT 3D facility.

   Table 1 lists the highest SNR observing opportunities for the three other terrestrial planets and the Moon during the period 2022-2032. The table lists the date of observation, SNR, estimated Doppler width, and range to the center of the target body.
We used the NASA HORIZON ephemeris (Giorgini et al., 2001) to find the elevation angle to each planetary body once per hour over the time period, and evaluated the closest pass for each body that was above the $30°$ cut-off elevation. We then calculated the SNR using the expression found in Ostro (1993); Kastinen et al. (2020), with a $25\%$ duty cycle and assuming a 0.1 radar albedo. While it is customary to report SNR obtained during the time it takes for the radar signal to make a round-trip to the target and back, we haved used SNR/hour, as the 25% duty-cycle allows for interleaved transmit and receive.



**Table 1.** Achievable SNR for hour-long observations of the terrestrial planets and the Moon. The date where the target has the best signal to noise ratio is shown, along with the Doppler width and approximate distance during the observation period.

| Target | SNR/hour | Minimum distance (m) | Doppler width (Hz) | Date (UTC) |
|--------|----------|----------------------|--------------------|------------|
| The Moon | 108 dB | $0.363 \cdot 10^9$ | 3.2 | 2022-Oct-10 23:00 |
| Venus | 28 dB | $4 \cdot 10^{10}$ | 3.2 | 2025-Mar-19 11:00 |
| Mars | 14 dB | $6 \cdot 10^{10}$ | 736 | 2022-Dec-01 00:00 |
| Mercury | 1 dB | $8 \cdot 10^{10}$ | 10.2 | 2028-Jun-02 08:00 |

It is obvious from table 1 that only the Moon is a viable radar target. The other terrestrial planets are simply too faint for scientific mapping purposes. While Venus and Mars may be detectable, they cannot be used to produce a range-Doppler radar image with very many pixels.

Target selection is limited by the achievable SNR. Due to the $R^{-4}$ dependence of the returned signal, distant objects quickly become lost in noise. In order to compensate for the low signal of distant objects, one can increase the effective collecting area of the radar receiver. It would therefore be possible that future expansions of the EISCAT 3D facility allows the study of more distant objects. In order to make a meaningful difference, the product of transmit power, gain, and receiver aperture would need to be increased by several orders of magnitude. Such an expansion may prove to be challenging in practice.

The small solar elongation angle to Mercury and Venus will probably increase the receiver noise significantly, as the Sun will often be relatively close to the radar antenna beam axis. Therefore, the SNR for Mercury and Venus are probably overestimated.

While the SNR calculations indicate that it may be possible to detect Venus, Mars, and Mercury with the E3D, they are not mappable radar targets. Assuming Venus has an SNR of 28 dB, this will only produce a radar map with approximately 8x8 pixels with an SNR of 10 dB per pixel. While it is possible to increase the SNR through increasing total integration time, this increase is approximately linear. The total amount of time required to obtain useful resolutions is so high as to not be worth attempting.

## 4 Range-Doppler disambiguation

The range-Doppler ambiguity can be seen in figure 2. Every point on the northern (positive Z"-direction) hemisphere is identifiable by one ring of constant range and one ring of constant Doppler shift. These rings also intersect on the southern hemisphere, assuming that the target is spherical. This means that any range-Doppler coordinate pair points to two physical locations, one north of the apparent Doppler equator, and one to the south. Untreated range-Doppler maps therefore appear to fold along the equator, adding together the regions on both hemispheres that have the same range and Doppler values. This folding makes it challenging to extract useful information from the maps.

The range-Doppler north-south ambiguity can be removed by an interferometric technique described by Rogers and Ingalls (1969). This method relies on using the phase difference between two receiving antennae to discriminate between echoes from the ambiguous points. While Rogers and Ingalls originally used only one interferometric antenna pair, the method can be





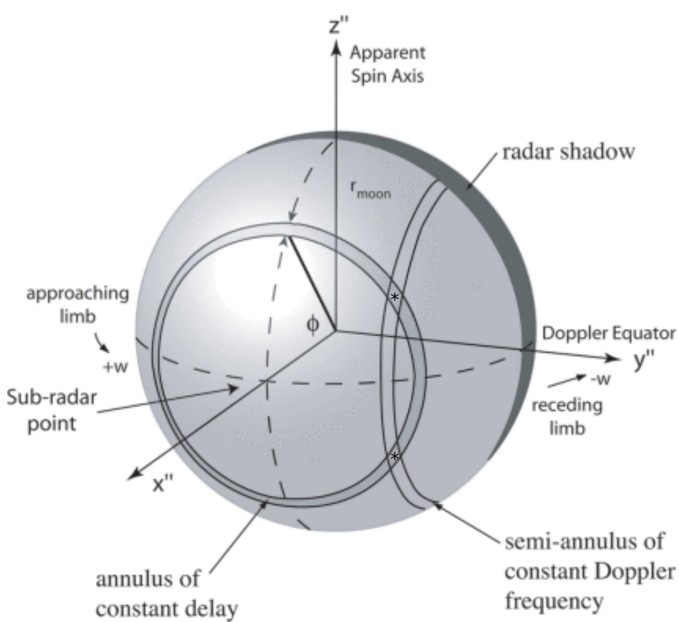

**Figure 2.** In this figure, the geometry of range-Doppler mapping is shown. Each range pixel takes the form of a ring on the surface of the Moon, centered on the sub-radar point. Each frequency bin takes the form of a ring, centered on the equator, 90 degrees east and west of the sub-radar point. Marked with a star are two regions that will have the same range and Doppler dimensions, symmetrical about the Doppler equator. These regions are indistinguishable with only range and Doppler information, and every point on the apparent north hemisphere will have a point on the south hemisphere with identical range and Doppler coordinates. Figure adapted from Campbell et al. (2007)

expanded to an arbitrary number of unique pairs. The addition of more antennae with different interferometric baselines makes the inverse problem of separating Doppler North from Doppler South more overdetermined. Discussing the interferometry in detail is out of the scope of this paper. We refer you to a companion paper by Stamm et al. (2020) for a discussion on the interferometric imaging capabilities of EISCAT 3D.

     When using a two antenna interferometer to disambiguate the Doppler North and South, there is a region near the Doppler
equator where the angular separation between the Doppler North and South region is very small. As the phase difference approaches zero near the Doppler equator, there will be a region surrounding it where the north-south ambiguity is unresolvable. The width of this gap to first order is inversely proportional to the separation of the antennas. An example of such a band is shown in figure 3. This map is made by Vierinen et al. (2017b) using data gathered by the Jicamarca radar facility, using a single interferometric antenna pair.

Due to the increased number of elements in the EISCAT 3D interferometer, as well as the availability of extremely long baselines, this ambiguous band will be significantly smaller. In combination with the regular and frequent observation opportunities, this means that EISCAT 3D could map the reflectivity of the low-latitude, low-longitude region of the lunar nearside.

     The data for the 2017 study by Vierinen et al. was collected with the Jicamarca radio observatory using the northernmost and southernmost modules, giving a baseline of 424 m. There are 64 modules, with side lengths of approximately 40 m each.

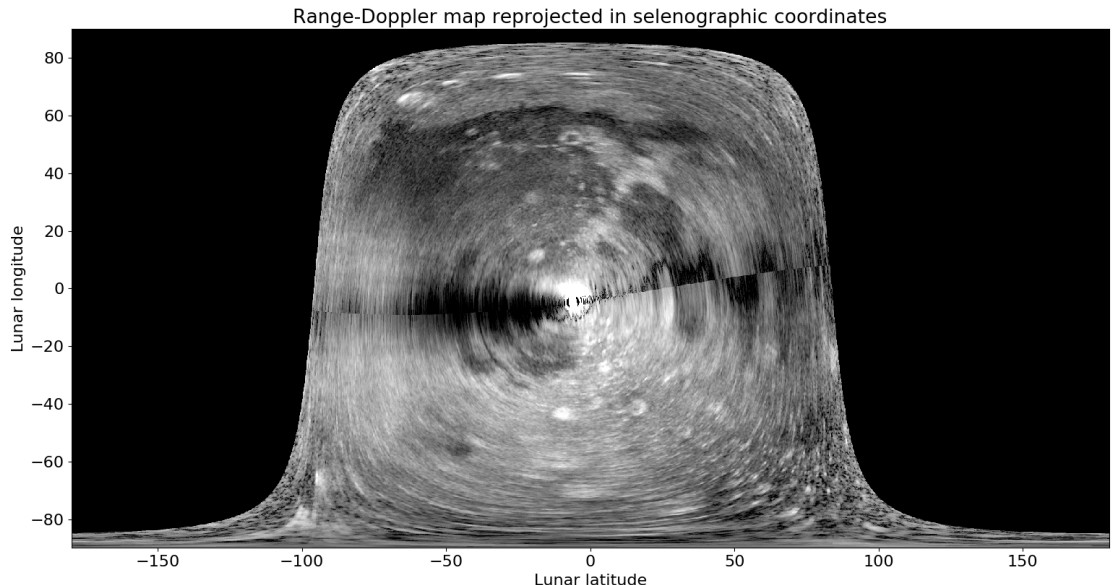

**Figure 3.** Due to uncertainties in the disambiguation calculation, a band surrounding the Doppler equator is poorly resolved. Note that the map is re-projected from range and Doppler coordinates to selenographic coordinates. Near the Doppler equator, features become lost in noise due to the noise-enhancing effects of poor disambiguation. This figure is produced from the same data as Vierinen et al. (2017b)

With a 6m wavelength, this gives an approximate gain of 27.5 dB at normal incidence. The EISCAT 3D outlier antennae are hexagonal with a maximal diameter of approximately 7.9 m with a wavelength of 1.3 m. We will assume that this also gives a gain of 27.5 dB at normal incidence. Due to the improved steering of the EISCAT facility over Jicamarca, targets are viewable at 30 degrees elevation from the horizon. This reduces the effective collecting area of the antenna, but also increases the possible observation time.

Interhemispheric cross-talk is the power from one hemisphere that is incorrectly measured as coming from the other hemisphere. In order to estimate the this for EISCAT 3D, we will be using the set of equations provided by Vierinen et al. (2017b), appendix 1. We will assume that the power only originates from one hemisphere, such that $p_n = 0$ and $p_s = 1$, meaning that all of the power measured is originating from the southern hemisphere. The estimate of interhemispheric cross-talk then becomes:

$$\mu = \frac{\sigma_n}{p_s} \tag{1}$$

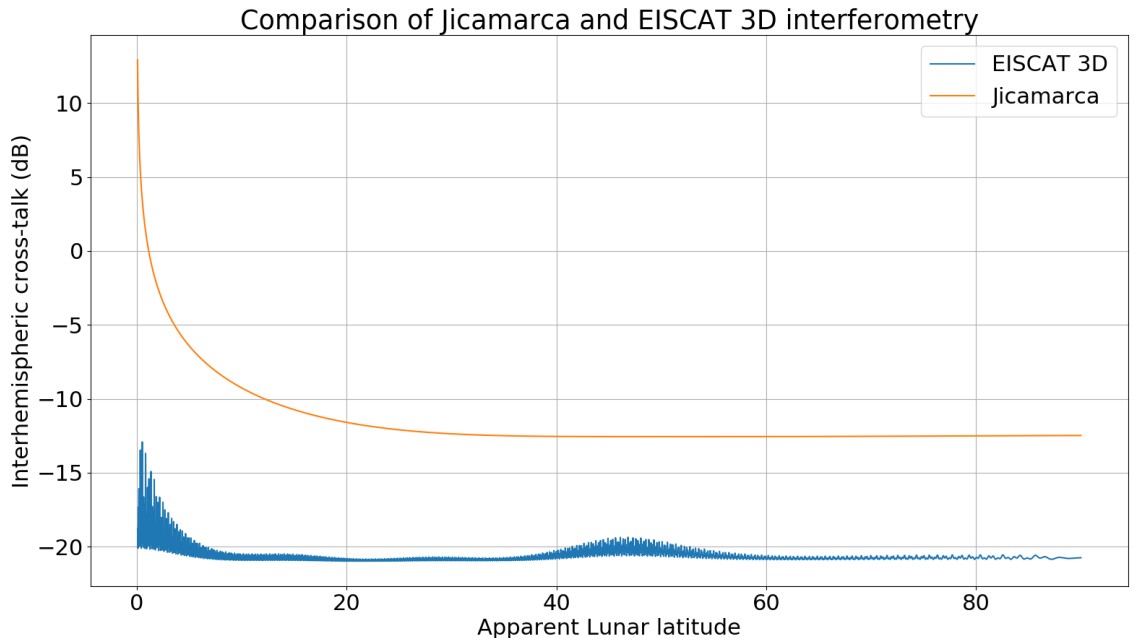

**Figure 4.** A comparison of interhemispheric crosstalk (equation 1) between the configuration used by Vierinen et al. (2017b) using the Jicamarca radio observatory and the EISCAT 3D facility using every unique interferometric baseline larger than 50 m. The apparent lunar latitude is terminated at approximately 0.05 degrees. Note that the estimate for EISCAT 3D is lower at all points than what is estimated for Jicamarca, which suggests that even regions close to the Doppler equator should be possible to disambiguate with the interferometric capabilities of EISCAT 3D.

Where $\sigma_n$ is the a posteriori standard deviation of the measured power from the northern hemisphere and $\mu$ is our estimate of interhemispheric cross-talk. A higher value for $\mu$ would mean that a larger amount of the power estimated to be originating from the northern hemisphere is actually from the southern hemisphere.

The reduction in gain at 30 degrees elevation is proportional to the reduction in projected area, leaving the total gain at
approximately 24 dB, which should be more than sufficient for lunar observations. As the outlier antennae have sufficient gain to act as interferometers, there are a total of 78 possible unique baselines. For our lunar calculations, we have assumed that the Doppler axis is aligned with Earth's, and excluded any baseline less than 50 m in the north-south direction to shorten calculation time.

In figure 4, we have evaluated the interferometric performance for EISCAT 3D and Jicamarca as a function of lunar latitude.
The interferometric performance is measured with interhemispheric cross-talk $\mu$, which is given in equation 1. From the figure, we can see that the EISCAT 3D achieves an interhemispheric cross-talk of approximately $-20$ dB. As a point of comparison, the Jicamarca Radio Observatory only attains $-12$ dB. This means that it is comparatively easier to distinguish between points





on the northern and southern hemispheres. This is due to the significantly larger number of interferometer baselines available with EISCAT 3D, making the linear regression problem of separating the Doppler North and South a highly overdetermined

problem. In the calculation of interhemispheric cross-talk, we have assumed that both EISCAT 3D and Jicamarca make 81 independent power measurements, as was done in the previous Jicamarca study (Vierinen et al., 2017b).

Therefore, as a consequence of the diversity of interferometer baselines, the ambiguous band will be significantly thinner with the EISCAT 3D radar than with the previous Jicamarca observation, allowing us to see regions quite near to the apparent equator. If observations are conducted some time apart, it is possible to sample the lunar surface with different sub-radar points

and apparent equators. If the apparent lunar rotation is in a different direction in two different maps, the unresolved band about the Doppler equator will fall in different regions of the Moon. The same is also true for maps with different sub-radar points. This means that if one is attempting to create a map with total coverage of the lunar nearside, a thin unresolved region means that fewer unique looks are required.

## 5   Ionospheric effects

The relatively long wavelength used by EISCAT 3D also provides a challenge in that it is affected more than shorter wavelengths by the Earth's ionosphere. Spatial variations in the plasma density causes spatial variations in the refractive index. The refractive index is found from the Appleton-Hartree equation which, for an unmagnetised, collisionless plasma, can be simplified to:

$$n^2 = 1 - \frac{f_p^2}{f^2}, \tag{2}$$

where $n$ is the refractive index, $f_p$ is the plasma frequency, and $f$ is the frequency of the electromagnetic wave. In our case, the radar frequency is 233 MHz, and typical ionospheric plasma frequencies will be between $1 - 10$ MHz. We can simplify the analysis by assuming a collisionless plasma since the majority of the electron density variations will be in the F-region where the electron collision frequency is negligible. The radar frequency is also sufficiently high that increased D-region electron density will not cause a significant fraction of the radio wave to be absorbed. Further, we can ignore the effects of the

magnetic field because we can use circularly polarized transmissions. In this case, the signal transmitted by the radar can be seen approximately as one of the two characteristic ionospheric propagation modes throughout the path from the radar to the Moon, which means that birefringent radio propagation effects do not play a major role.

Irregularities in the electron density of the ionosphere are relatively common. As a consequence of electron density variations, signals traveling through the ionosphere will have small differences in phase due to variations of the optical path length.

This means that features which should be sharp in the range-Doppler map become blurred in the Doppler dimension. This effect can be partially counteracted with background knowledge of target features or knowledge of the ionospheric electron content. If there is a feature that is known to be a sharp and point-like (e.g. a crater rim smaller than the resolution cell or the sub-radar point), one can use this fact to estimate the effect of the phase modulation, and remove it or at reduce its effect, as was done in the Jicamarca study (Vierinen et al., 2017b).





Traveling Ionospheric Disturbances (TIDs) are wave-like features propagating in the ionosphere bringing enhancements and reductions to the background electron density. These disturbances can be caused by gravity waves propagating in the neutral thermosphere. Usually TIDs propagate perpendicular to their phase fronts (Shiokawa et al., 2013). The phase plane can be aligned vertically or tilted. They tend to have long wavelengths, down to approximately one hundred kilometers in the spatial scale and 15 minutes in the temporal scale. The buoyancy period at ionospheric heights is approximately 9 minutes and

hence non-evanescent gravity wave driven TIDs with considerable amplitudes will have a period larger than or equal to this. The amplitude of TIDs can vary from 0-15% (Bolmgren et al., 2020). Over Northern Scandinavia, it is found that the TIDs occur predominantly during pre-midnight hours 18 - 24 LT and their occurrence is scarce during post-midnight hours 0 - 6 LT (Shiokawa et al., 2013).

The variability of the electron density alters the observed signal most severely if the radar look direction is perpendicular

to the normal vector of the TID phase front. The right hand side of Figure 5 depicts this case. Then the variation of the total electron content along the beam path will be higher than if the signal propagates through multiple TID undulations, as shown on the left hand side of the same Figure.

In order to estimate the effect of TIDs for inverse synthetic aperture radar measurements, we have used a simple toy model for a monochromatic TID. Our electron density model is based on a background electron density profile $N_0(z)$, on top of which

a TID is overlayed on. In this case, the electron density in the x-z plane is given by the following equation:

$$N_e = [1 + \alpha \cos(k_z z + k_x x - 2\pi f_{TID} t)] N_0(z). \tag{3}$$

Here $k_x$ and $k_z$ are the horizontal and vertical wavenumbers of the TID, and $f_{TID}$ is the temporal frequency of the TID. The parameter $\alpha$ determines the amplitude of the fluctuating component of the electron density. Variables $x$, $z$, and $t$ denote the horizontal, vertical, and temporal dimensions.

The ionospheric contribution to the phase (in radians) of a radio signal of frequency $f_{\text{rad}}$ travelling through a plasma can be written as (Davies, 1965; Vierinen, 2011):

$$\phi = \frac{e^2}{2\pi \epsilon_0 m_e c f_{\text{rad}}} \int_L N_e d\boldsymbol{\ell} \tag{4}$$

here $L$ is the path of the signal, $e$ is the charge of an electron, $\epsilon_0$ is the permittivity of free space, $m_e$ is the electron rest mass, $c$ is the speed of light in vacuum, $f_{\text{rad}}$ is the frequency of the radio wave, and $N_e$ is the electron density. Note that we have

assumed a round-trip propagation of the radio wave.

By combining equations 3 and 4, we get an explicit expression for the TID effect on the phase of the received signal:

$$\phi(t) = \frac{e^2}{2\pi \epsilon_0 m_e c f_{\text{rad}}} \int_L N_0(\ell_z)[1 + \alpha \cos(k_z \ell_z + k_x \ell_x - 2\pi f_{TID} t)] d\boldsymbol{\ell} \tag{5}$$

The x and z components of the path evaluated in the integral are given by $\ell_x$ and $\ell_z$.





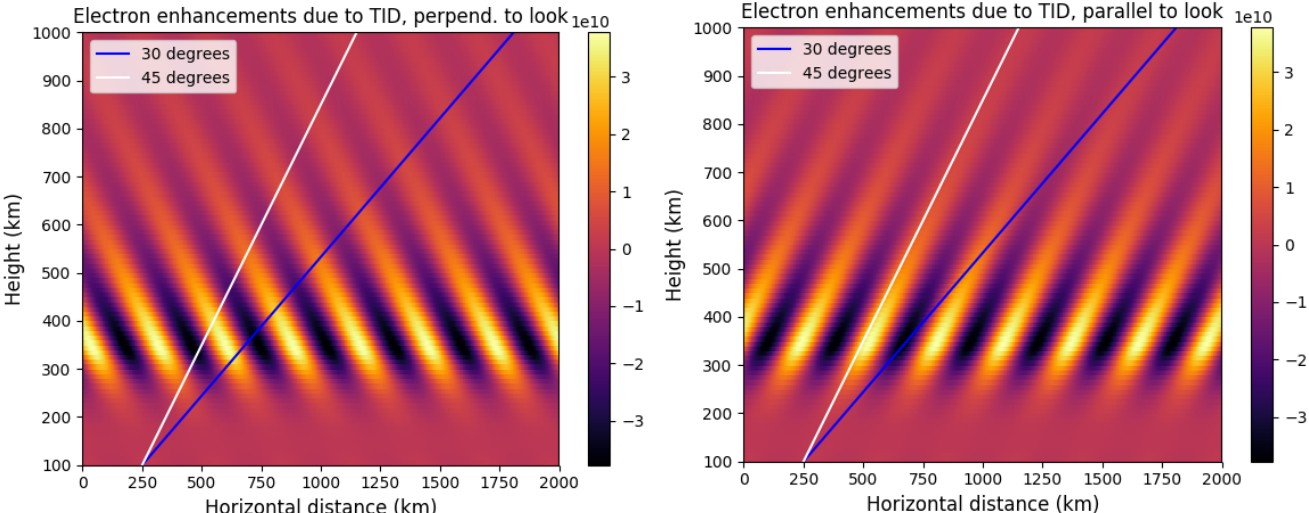

**Figure 5.** Model of electron enhancements due to TIDs. *Left:* The TIDs have been tilted 45 degrees to the left. Radar look directions for 45 and 30 degrees above the horizon is shown. *Right:* the same as the plot on the left, but with the TIDs tilted 45 degrees right. Both are in units of electrons per cubic meter.

By differentiating equation 5 with respect to time, we get the frequency shift (Doppler shift due to time-variable ionospheric radio propagation) of the signal:

$$\frac{d\phi(t)}{dt} = \frac{e^2 \alpha f_{TID}}{\epsilon_0 m_e c f_{rad}} \int\limits_L N_e(\ell_z) \sin(k_x \ell_x + k_z \ell_z - 2\pi f_{TID} t) d\boldsymbol{\ell} \tag{6}$$

where $\alpha$ is the TID electron density enhancement magnitude, $f_{TID}$ is the frequency of the TID in hertz, $k$ is the wavenumber of the TID, $f_{rad}$ is the frequency of the radar in hertz, $\boldsymbol{l}$ is a vector element along the path $L$.

This equation provides the round-trip rate of change of phase as a function of time due to electron density variations in both space and time, in units of radians per second. We evaluate this over an hour and obtain the minimum and maximum Doppler shift due to ionospheric radio propagation. We will call this effect ionospheric Doppler broadening.

In order to evaluate ionospheric blurring of inverse synthetic aperture radar images, we have evaluated different relative look angles between the TID phase front and the radio wave propagation direction. Angles between $0°$ and $90°$ with $15°$ were considered. Figure 5 shows the fluctuating component of the electron density associated with the TID model for different look angles. The left panel of figure 5 shows the best-case scenario, where the radar look direction is perpendicular to the phase front ($90°$), and the right panel shows the worst-case scenario, where they are parallel ($0°$). We have chosen to use an elevation angle of $45°$ for all cases, as this is close to the value of the highest elevation angle that the Moon can be observed using the EISCAT 3D radar. For the model calulations, we used TIDs with a wavelength of 200 km with a period of 10 minutes. The amplitudes are varied from $1\%$ to $10\%$ ($\alpha = 0.01$ to $\alpha = 0.1$. For all model calculations, we have assumed a night time electron density profile $N_0(z)$ based in the International Reference Ionosphere (Bilitza, 2001). We have scaled this electron density profile by





**Table 2.** Effect of TIDs on Doppler broadening under various conditions.

| $\alpha$ | TECU | Doppler broadening | % of lunar Doppler width |
|:---:|:---:|:---:|:---:|
| 0.01 | 10 | 0.03 Hz | 1% |
| 0.05 | 10 | 0.10 Hz | 3% |
| 0.1 | 10 | 0.34 Hz | 11% |
| 0.01 | 40 | 0.14 Hz | 4% |
| 0.05 | 40 | 0.69 Hz | 21% |
| 0.1 | 40 | 1.38 Hz | 43% |

**Table 3.** Effect of the angle between the phase plane and the radar look direction on Doppler broadening, with $\alpha = 0.1$ and TECU $= 40$.

| Relative look angle (Deg) | Doppler broadening (Hz) | % of lunar Doppler width |
|:---:|:---:|:---:|
| 0 | 1.38 | 43.0% |
| 15 | 0.56 | 17.5% |
| 30 | 0.22 | 6.8% |
| 45 | 0.10 | 3.2% |
| 60 | 0.06 | 1.8% |
| 75 | 0.04 | 1.2% |
| 90 | 0.03 | 0.9% |

a constant value in order to obtain a certain vertical total electron content:

$$\text{TEC} = \int_L N_e d\ell, \tag{7}$$

where $L$ is a vertical path through the ionosphere. We have evaluated the model for vertical TEC of 10 and 40 with units of $10^{16}$ electrons per square meter (TECu), corresponding to a low and high ionospheric electron density.

We find that the Doppler broadening depends on: 1) the amplitude of TEC enhancement, 2) the background TEC value, and 3) the relative look angle. This is expected since the a relative modulation will have a larger effect on the absolute total variation when the background density is higher. With larger look angles, the effect of ionospheric electron density fluctuations average out more and result in smaller ionospheric phase variations.

In table 2, we have compiled estimates of the ionospheric Doppler broadening caused by a TID where the radar look
direction is in the phase plane, which corresponds to the worst case scenario (look angle $0°$). We have assumed a Lunar Doppler width of 3.2 Hz, which would be a typical Doppler width for observations in 2022. The ionospheric Doppler broadening is approximately linear with both TEC and TID electron enhancement amplitude.

In table 3 we have compiled estimates of frequency distortions of a two-dimensional plane-wave TID with an electron enhancement of $10\%$ on a background ionosphere of approximately 40 TECU. In this simulation, we have assumed a radar
elevation angle of $45°$. In order to evaluate the impact of changes in the angle between the TID phase plane and the radar look





angle, we changed the tilt of the TID phase plane from what is shown in the left panel of figure 5 to the right panel in $15°$ steps. We can see that the angle to the phase plane is of critical importance for the effect of the phase disruption. When the signal travels through multiple waves, the time dependence of the total electron content is drastically reduced.

The Doppler broadening effects of TIDs are highly variable, and can go from negligible to almost as large as the rotational Doppler shift. This is effect is largest when the angle between the radar look direction and the TID phase plane becomes small. These simple model calculations are representative of what can be expected. Further, previous studies of ionospheric TIDs from Tromsø indicate that conducting the experiment during post midnight hours in geomagnetically quiet periods is preferable in order to reduce the Doppler broadening effects caused by the ionosphere. Moreover, EISCAT 3D can be used to accurately measure the ionospheric electron density during observations, which can then be used to correct for phase variations caused by
the ionosphere.

## 6   Future lunar observation opportunities

In order to aid the planning of future Lunar observations with EISCAT 3D, we have compiled the possible observation opportunities of the lunar face from the year 2022 to the year 2040. Observation opportunities are plentiful, but vary significantly from year to year when it comes to the location of the sub-radar point, Doppler width, and the orientation of the Doppler equator.

Figure 6 is a plot of the lunar elevation in ten minute increments in the year 2022, which is when the EISCAT 3D facility is scheduled for completion. As the facility can steer $60°$ off zenith, there are several days every month suitable for lunar observation. Note that each spike consists of several days (~10), with observation opportunities several hours long for most days. The elevation charts of other years are relatively similar. The maximum elevation varies slightly as the orbit of the Moon varies, but this variation does not significantly impact the frequency or duration of observation opportunities.

Figures 7 and 8 further show the sub-radar point in selenographic coordinates from the year 2022 up to year 2040. The line is green when the lunar face is at least 30 degrees over the horizon, as seen from EISCAT 3D in Skibotn. Over this period, the sub-radar point migrates around the origin of the selenographic coordinate system. Due to this migration, the view of the lunar nearside changes as the horizons and rotation direction change. The mid 2020s will provide a view of the lunar south pole, while the mid 2030s will allow us to study the northern regions. As the sub-radar point migrates day-to-day as well as
year-to-year, some regions near the lunar limbs are only visible a fraction of the time. Care should then be taken to ensure that interesting regions are observed when they are observable, as opportunities do not repeat often.

## 7   Conclusions

EISCAT 3D provides an excellent new tool for radar imaging of the lunar nearside. Observation opportunities are plentiful and varied, and the radar can track the lunar face for sufficiently long periods of time to allow high quality observations to be made.
The operating frequency of EISCAT 3D is previously unused for lunar mapping purposes, and will therefore provide new information about the scattering properties of the lunar terrain. The interferometric capabilities provided by the large number

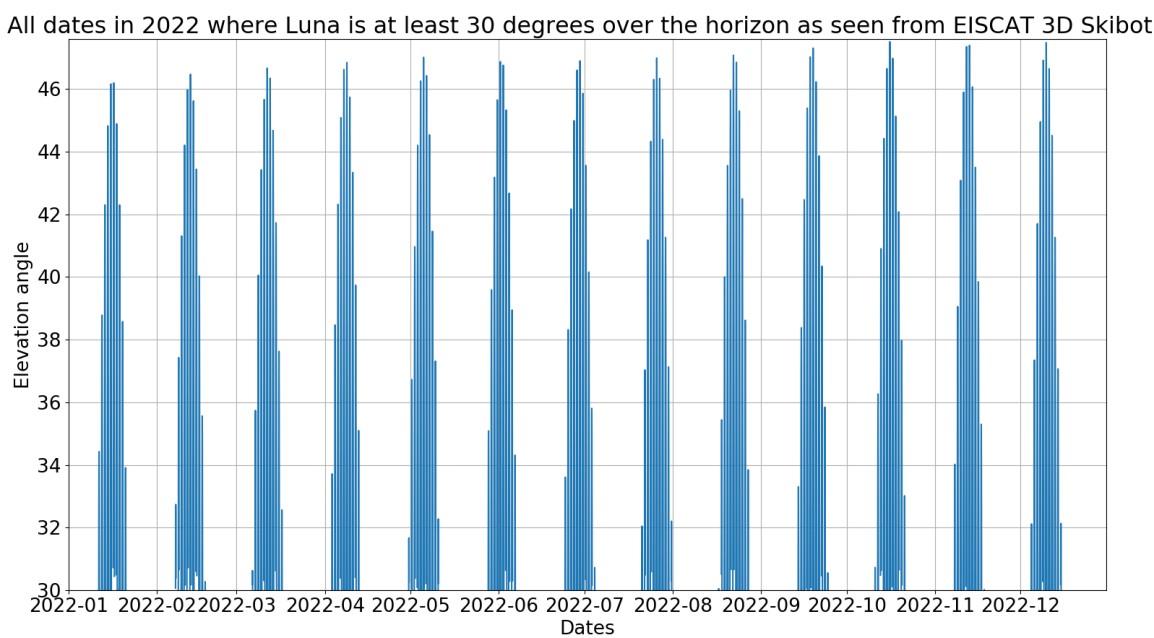

**Figure 6.** All dates where the lunar face is more than 30 degrees over the horizon during 2022, as viewed from the EISCAT 3D Skibotn site. Note that each spike is a lunar month, and every lunar month has several days where observations are possible.

of receiving antennae are able to resolve the north-south ambiguity, even close to the Doppler equator. This means that it will take fewer unique looks to obtain a full coverage map of the lunar nearside.

The other terrestrial planets are too far away for EISCAT 3D to achieve a scientifically useful resolution due to the low signal
strength. This could possibly be rectified by increasing the transmitted power and/or receiver gain function of the facility some time in the future, though this may be unrealistic in practice.

The ability of the radar facility to track moving targets and steer down to $30°$ elevation allows for many long observation opportunities. This will be useful for mitigating the disrupting effects of ionospheric variability, as experiments can easily be rescheduled. TIDs can be a significant hindrance to obtaining clear radar images, but the most disruptive events should not be
very common.

The variability of the lunar sub-radar point and Doppler axis will allow for varied views of the lunar face. Regular observations over the course of several years can provide new views of regions that are occasionally obscured by the lunar horizon.

This article provides an expansion of the science case for EISCAT 3D into the realm of planetary radar mapping. While Near-Earth Objects can also fall under the umbrella term of planetary radar, this science case is discussed in detail in the
companion paper by Kastinen et al. (2020), which will appear in the same special issue.



**Figure 7.** Lunar observation opportunities 2022-2030. The position of the subradar point in selenographic coordinates is shown in light gray. All subradar point positions for which the Moon is above $30°$ elevation are shown in green. Each year offers a slightly different view of the Moon.

**Figure 8.** Lunar observation opportunities 2031-2039. This figure is the same as figure 6, but extended further into the 2030s.



*Author contributions.* Torbjørn Tveito performed numerical calculations for TID simulations, interferometric disambiguation, and iono-spheric frequency disturbances. Juha Vierinen performed the compilation of observation opportunities and signal-to-noise ratio calculations of planetary targets. Björn Gustavsson contributed to the calculation for observation opportunities, interferometric disambiguation, and TID simmulation. Viswanathan Lakshmi Narayanan contributed to section 5, for both the calculation and interpretation of results. All authors contributed to the writing of the manuscript and the interpretation of results.


*Competing interests.* Juha Vierinen is on the editorial board of this special issue

*Acknowledgements.* Acknowledgements. T. Tveito and J. Vierinen would like to thank the Tromsø Research Foundation for supporting this work.



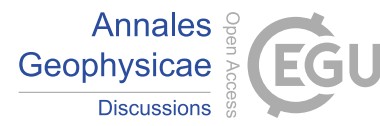

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
