# Peer review of "Planetary Radar Science Case for EISCAT 3D"

_Annales Geophysicae, 2020_

## Referee Comment (RC1) · Sriram Bhiravarasu (Referee) · 22 Dec 2020

As demonstrated in this well written paper, EISCAT 3D would provide excellent opportunities for lunar mapping at a wavelength that was never used before to understand the physical properties of deep lunar subsurface. Important aspects in ground-based radar mapping which are the ionospheric Doppler broadening and North-South ambiguity are well addressed and discussed in detail in this paper. As a planetary radar user, I am particularly interested to see the results of the planned lunar mapping campaigns described in the last section. I don't have any major concerns with the text and figures presented here, so I would strongly recommend this paper for publication. However, I have a few minor suggestions (below) which could be addressed by the authors before this manuscript gets published:

1) Line 31: Stacy et al., 1997 is not a good reference to use when you describe about

the radar studies of water ice for solar system objects. This study particularly describes the lunar polar observations using Arecibo radar. Campbell et al., 2016 (used in this paper) might be a good alternative here. 2) Line 33: Spudis et al., 2013 didn't use Arecibo data in their analysis. The results of this paper is based exclusively on LRO Mini-RF data. So the phrase "in conjunction with Arecibo radar" may be removed. 3) Line 38: "... that the planet was geologically active relatively recently..." : Add a reference here 4) Line 43: "Compositional" might be a better word here instead of "chemical" 5) Line 59-60: Rephrase the sentence as "Radar observations of Asteroids can also aid in determining their spin state through...." for the Busch et al., 2010 reference. 6) Line 112: Replace "very many" with "sufficient" 7) Lines 133- 135: It would help the reader if the beamwidth of EISCAT 3D can be mentioned here in case of lunar mapping. Does the EISCAT 3D beamwidth encompasses the entire lunar globe for a single "snapshot"? Also, is it possible to point the radar beam to the desired hemisphere (N/S) by slightly adjusting the radar beam as done in the case of Venus mapping from the Arecibo radar? For Arecibo Venus mapping, see Campbell, 2002 and Campbell et al., 2016. 8) Line 195: If you employ circularly polarized Tx only, how can you generate the full scattering matrix as mentioned in the abstract and section 2? And when you employ linear pol for Tx, how would you compensate for the Faraday effect? A small note on this discussion may be included here. 9) Line 245: Does the left panel of figure 5 indicate the radar look direction as perpendicular to the phase front or parallel? Because it was mentioned as parallel in line 214. 10) Lines 271-273: Add a reference here for previous ionosphere studies at Tromso region.

---

## Referee Comment (RC2) · Anonymous Referee #2 · 31 Dec 2020

The present manuscript describes the capabilities of the new EISCAT 3D radar to perform planetary radar science. In particular, the manuscript explains that lunar mapping is a plausible application for the EISCAT 3D radar and that there would be multiple opportunities to conduct this type of observations in the following years. The document conducts a comparison between expected results for the EISCAT 3D radar and lunar measurements conducted with the Jicamarca radar in recent years, the comparison shows that EISCAT 3D images would be of higher quality providing useful information to study the characteristic of Moon surface. Since the proposed technique would generate lunar observations with a radar wavelength not used before for this type of studies, the results obtained with EISCAT 3D would complement previous studies and observations. Since the document is also well written and organized I would recommend its publication after the following minor comments are addressed. * In section 3, I would recommend to include the expression used to estimate the SNR of plane-

tary targets and the parameters used in its calculation in order to be able to reproduce Table 1. This would facilitate the interpretation of the results presented here. * In section 4, I would suggest to compute and discuss the expected resolution of the lunar radar images to be obtained with EISCAT 3D. These values should be compared with the Jicamarca radar observations in order to discuss the improvement that would be achieved using the EISCAT 3D radar. * Given that lunar echo signals would be obtained at low elevation angles (30 degrees) using the EISCAT 3D radar, it is likely that the shape of the antenna beam pattern would have an impact on the observations distorting the reconstructed images. I would suggest the authors to consider including a discussion about this in the manuscript.

---

## Author Comment (AC1) · 28 Jan 2021

In this document, I will present the comments from Sriram Bhiravarasu in bold preceded by a right chevron

>**As such.**

My reply will be in plain text.

I would like to begin by thanking Sriram Bhiravarasu for his effort in reviewing the manuscript.

>**As demonstrated in this well written paper, EISCAT 3D would provide excellent opportunities for lunar mapping at a wavelength that was never used before to understand the physical properties of deep lunar subsurface. Important aspects in ground-based radar mapping which are the ionospheric Doppler broadening and North-South ambiguity are well addressed and discussed in detail in this paper. As a planetary radar user, I am particularly interested to see the results of the planned lunar mapping campaigns described in the last section. I don't have any major concerns with the text and figures presented here, so I would strongly recommend this paper for publication. However, I have a few minor suggestions (below) which could be addressed by the authors before this manuscript gets published:**

>**1) Line 31: Stacy et al., 1997 is not a good reference to use when you describe about the radar studies of water ice for solar system objects. This study particularly describes the lunar polar observations using Arecibo radar. Campbell et al., 2016 (used in this paper) might be a good alternative here.**

>**2) Line 33: Spudis et al., 2013 didn't use Arecibo data in their analysis. The results of this paper is based exclusively on LRO Mini-RF data. So the phrase "in conjunction with Arecibo radar" may be removed.**

>**3) Line 38: "... that the planet was geologically active relatively recently..." : Add a reference here**

>**4) Line 43: "Compositional" might be a better word here instead of "chemical"**

>**5) Line 59-60: Rephrase the sentence as "Radar observations of Asteroids can also aid in determining their spin state through...." for the Busch et al., 2010 reference.**

>**6) Line 112: Replace "very many" with "sufficient"**

I agree with points 1 through 6. The reviewers comments are well founded and agreeable. The recommended changes have been added to the manuscript.

>**7) Lines 133- 135: It would help the reader if the beamwidth of EISCAT 3D can be mentioned here in case of lunar mapping. Does the EISCAT 3D beamwidth encompasses the entire lunar globe for a single "snapshot"? Also, is it possible to point the radar beam to the desired hemisphere (N/S) by slightly adjusting the radar beam as done in the case of Venus mapping from the Arecibo radar? For Arecibo Venus mapping, see Campbell, 2002 and Campbell et al., 2016.**

The beamwidth of EISCAT 3D varies with elevation angle. At 90 degrees, the half-power beamwidth is approximately one degree (two moons). As the radar is steered towards 30 degrees, the beamwidth becomes elongated to approximately five degrees (ten moons) wide. This makes it impractical to attempt to illuminate only one hemisphere at a time. The difference in illuminating power from one hemisphere to the other will not be large enough to be able to ignore the ambiguity. The interferometric disambiguation method will be a more reliable tool for separating the northern and southern hemispheres.

>8) Line 195: If you employ circularly polarized Tx only, how can you generate the full scattering matrix as mentioned in the abstract and section 2? And when you employ linear pol for Tx, how would you compensate for the Faraday effect? A small note on this discussion may be included here.

A set of two linearly independent polarizations can be used to synthesize any polarization state. One way of estimating the two-way Faraday rotation is by using the sub-radar point. We can assume that this area near the sub-radar point is "flat" and aligned normal to the propagation direction, and will therefore act like a mirror. The signal we measure from the sub-radar point can then be assumed to be dominated by specular scattering, including the effects of two-way Faraday rotation. As such, the polarization of the signal from the sub-radar point will then be the polarization we expect from specular scattering from the entire lunar face. It follows that the polarization expected from subsurface scattering is the polarization which is linearly independent from the one dominating the sub-radar point.

>9) Line 245: Does the left panel of figure 5 indicate the radar look direction as perpendicular to the phase front or parallel? Because it was mentioned as parallel in line 214.

These waves are transverse waves, and so the wave vector is always perpendicular to the phase front. In line 214, I am discussing the TID wave vector, while in line 245 i am discussing the phase front. This phrasing can easily confuse readers, and one way to rectify it may be to add an arrow in figure 5 showing the wave vector, and adding that these are transverse waves. See figure 1 for a suggested change to figure 5 (only showing the left part). The cyan arrow will be described as the wave vector of the TID.

>10) Lines 271-273: Add a reference here for previous ionosphere studies at Tromso region.

The book Physics of the Upper Polar Atmosphere by Asgeir Brekke might be a suitable reference here. Many of the figures in the book are based on studies from the region around Tromsø.

[Figure]

Figure 1: Updated version of figure 5 with wave vector

---

## Author Comment (AC2) · 28 Jan 2021

In this document, I will present the comments from reviewer 2 in bold preceded by a right chevron

>**As such.**

My reply will be in plain text.

I would like to begin by thanking the anonymous reviewer for their effort in reviewing the manuscript.

>**The present manuscript describes the capabilities of the new EISCAT 3D radar to perform planetary radar science. In particular, the manuscript explains that lunar mapping is a plausible application for the EISCAT 3D radar and that there would be multiple opportunities to conduct this type of observations in the following years. The document conducts a comparison between expected results for the EISCAT 3D radar and lunar measurements conducted with the Jicamarca radar in recent years, the comparison shows that EISCAT 3D images would be of higher quality providing useful information to study the characteristic of Moon surface. Since the proposed technique would generate lunar observations with a radar wavelength not used before for this type of studies, the results obtained with EISCAT 3D would complement previous studies and observations. Since the document is also well written and organized I would recommend its publication after the following minor comments are addressed.**

>**In section 3, I would recommend to include the expression used to estimate the SNR of planetary targets and the parameters used in its calculation in order to be able to reproduce Table 1. This would facilitate the interpretation of the results presented here.**

The SNR of planetary targets was found using the same method as in the companion paper, Radar observability of near-Earth objects using EISCAT 3D (Kastinen et al.),

The expression and a short description is added to the manuscript. These equations are discussed in further detail in Planetary Radar Astronomy (Ostro, 1993) and the companion paper.

>**In section 4, I would suggest to compute and discuss the expected resolution of the lunar radar images to be obtained with EISCAT 3D. These values should be compared with the Jicamarca radar observations in order to discuss the improvement that would be achieved using the EISCAT 3D radar.**

I have added a paragraph describing the theoretically achievable resolution of a range-Doppler map using EISCAT 3D. The range resolution $R_r$ achievable is found as $R_r = \frac{c}{2B}$, where $B$ is the transmit bandwidth and $c$ is the speed of light. As EISCAT 3D has a transmit bandwidth of 5 MHz, this results in a range resolution along-sight of approximately 30 m. The frequency resolution $R_f$ along the equator is found as $R_f = \frac{\lambda D_m}{2 v_{rot} c \tau_m}$, where $D_m$ is the diameter of the Moon, $v_{rot}$, and $\tau_m$ is the observation time in seconds. The apparent rotation velocity of the Moon will change day-to-day, but will be be somewhere between 0.5 m/s and 2.0 m/s For a one-hour observation with an apparent rotation velocity of 1.2 m/s, the average resolution in the Doppler dimension

will be 520 m

The practically achievable resolution will be significantly lower than what is theoretically possible. Much of this reduction comes from efforts to compensate for low SNR and to reduce speckling. Another challenge is that Rogers and Ingalls method of north-south disambiguation assumes a stationary Doppler axis. This assumption does not hold for long observations of the lunar face, effectively limiting possible observation times. These effects will be dependent upon the specifics of each observation, and can be expected to vary significantly.

>**Given that lunar echo signals would be obtained at low elevation angles (30 degrees) using the EISCAT 3D radar, it is likely that the shape of the antenna beam pattern would have an impact on the observations distorting the reconstructed images. I would suggest the authors to consider including a discussion about this in the manuscript**

The relatively low elevation angle of the lunar face presents three challenges for radar imaging. 1) The antenna gain pattern will be elongated in the elevation direction. This will cause a loss of signal strength. 2) The elevation pointing direction will also cause a polarization dependent phase and amplitude response for the antenna. This will require careful calibration. 3) The point spread function of the interferometer will also be affected.

Of these effects, the polarization dependent antenna response is probably most important. This will most likely require the community to develop an azimuth and elevation dependent polarization response model for the EISCAT 3D antenna.